# Social media research: We are publishing more but with weak influence

**Samer Elhajjar[1], Laurent Yacoub** [2]*

**1** Department of Marketing, National University of Singapore, Singapore, Singapore, **2** Department of Business Administration, Holy Spirit University of Kaslik, Jounieh, Lebanon

* laurent.yacoub@usek.edu.lb

## Abstract

The purpose of this paper is to address the chasm between academic research on social media as an expanding academic discipline and at the same time a growing marketing function. A bibliometric analysis indicated the evolution of academic research on social media. The results of a survey of 280 social media practitioners shed the light on the gap between academic social media research and the practice of professionals. A qualitative study also offered novel insights and recommendations for future developments in academic research on social media. The findings of this paper showed that academic research on social media is growing in terms of the number of publications but is struggling in three areas: visibility, relevance, and influence on practitioners. This study contributes to the body of knowledge on social media. The implications of our study are derived from the importance of our findings on the directions to publish more relevant and timely academic research on social media. While extensive studies exist on social media, their influence on practitioners is still limited.

**Data Availability Statement:** All relevant data are within the manuscript and its Supporting Information files. The data underlying the results presented in the study are from Scopus (http://www.scopus.com/).

## Introduction

In the early 1960s, academics began to advocate that marketing should gain rigor by relying on a scientific approach that respects requirements in terms of the state of knowledge, the hypotheses development, the methodology, and the analysis and interpretation of results [1–5]. This traditional conception of rigor has, over the years, fuelled the need to acquire tools to better evaluate, recognise and promote it. Thus, first in the United States, and now in almost all countries, various stakeholders use rankings of scientific journals, mainly Anglo-Saxon, which often consider their impact factors according to the Journal Citation Report (JCR) of the Science Citation Index or the Social Science Citation Index. Quality accreditation bodies for higher education management institutions have also followed suit by offering journal ranking lists.

However, since the early 1980s, the debate between rigor and relevance in the production and dissemination of marketing knowledge has been prominent in the literature. There is a serious concern about how academics are evaluating the impact of their research. As if the focus of marketing researchers is to improve their citation records rather than developing

**Funding:** The author received no specific funding for this work.

**Competing interests:** The authors have declared that no competing interests exist.

practical implications for practitioners. Shouldn't marketing scholarship, when applied to practical issues, aim to harmonize rigor and relevance right from the start? How did we arrive at this risk of divorce and the need to reconcile thoroughness and applicability?

In fact, in some fields, such as pharmacy, where breakthroughs in medical procedures and the discovery of new pharmaceuticals result in societal benefits, the influence of research is simple to grasp. This effect is more difficult to detect in social media. In the discipline of marketing, for example, there have been allegations that research has strayed too far from the interests of practitioners. In turn, researchers point out the flaws in present professional methods [6]. Indeed, some in the marketing research community believe that many practical concerns that worry professional marketers are unworthy of researchers' attention. This is mainly because of a long-standing misguidance of business schools [7] since their research is less and less influential [8–11]. Several studies confirm that the impact of academic research on business practices has been disappointing and that innovations have come from the consulting community, the business press, and professional associations [12–15].

This article aimed to identify whether there is a gap between rigor and relevance in academic research on social media. It also proposed ways for marketing researchers to foster relevance. In general, this article responded to two research questions: Is there a chasm between academic social media research and social media practitioners? How to reconcile the rigor and relevance of social media research?

The originality of this research lies in its specific focus on bridging the potential gap between rigor and relevance within the realm of academic research on social media. While social media has become an integral part of contemporary society and communication [16], there is a growing concern that academic investigations in this domain may sometimes prioritize theoretical rigor at the expense of practical applicability [17]. By addressing this issue, the research seeks to contribute significantly to the field by shedding light on the balance between rigorous methodologies and the real-world applicability of social media research findings. This unique perspective not only emphasizes the importance of ensuring academic work remains pertinent and useful in a rapidly evolving digital landscape but also offers valuable insights for researchers, educators, and policymakers striving to navigate the intricate intersection of academia and social media's dynamic environment.

To answer our research questions, the paper was structured as follows. First, we examined the theoretical foundations of academic marketing research. Second, the research design and methodology of our three investigations were then described. Our first study involved a social media research bibliometric analysis with the goal of describing the evolution and development of academic social media research. Our second study gathered feedback and information from social media practitioners. Our third study listed suggestions for academic researchers. The three studies worked in tandem to create a comprehensive picture of academic research on social media. They offered historical context, practical insights, and actionable recommendations, collectively contributing to a holistic understanding of how researchers can bridge the gap between rigor and relevance in the dynamic realm of social media. Lastly, we listed the contributions of our study and their implications for future research.

## Literature review

Academic marketing research has two purposes: first, to advance marketing theory, and second, to improve marketing practice [18]. On the one hand, theory ought to give academics fresh ideas, conceptual frameworks, and resources to aid in their understanding of marketing phenomena. On the other side, research should give marketers direction for making better decisions. As a result, marketing academics should address issues with the development of

marketing theory's rigor and its applicability to marketing practice [19]. Nevertheless, leading academic voices have expressed worry about the gap between marketing theory and practice. Reibstein et al. [20], for example, have questioned if marketing academia has lost its way, while Sheth and Sisodia [21] have urged for a reform. In a similar vein, Hunt [22] advised revising both marketing's discipline and practice, while McCole [23] proposed strategies to refocus marketing theory on changing practice. Rust et al. [24] argue for reorienting marketing in firms to become more customer-centric, and Kotler [25] advocates for marketing theory and practice to conform to environmental imperatives. Also, because the business landscape is dynamic, Webster Jr. and Lusch [26] believe that marketing's goal, premises, and models should be rethought. Finding answers to these problems keeps marketing from becoming obsolete [27] and marginalised [20], both as a discipline and as an organizational function [28].

According to the literature [29, 30], marketing scholars have lost sight of both rigor and relevance. As a result, many scholarly journals have made it normal practice to provide implications and suggestions [31], their actual influence has been insignificant [20]. Many marketing academics have failed to address substantive topics [18], resulting in a loss of relevance [30, 20] and a drop in marketing expertise [32].

The efforts of certain institutions (e.g., Marketing Science Institute), conferences (e.g., Theory + Practice in Marketing–TPM), and leading journals' special issues on marketing theory and practice to bridge this gap are well recognised, with the goal of fostering dialogue and collaboration between marketing scholars and practitioners. Several solutions for bridging the marketing theory–practice gap have also emerged from existing literature: adopting the perspective of rigor–and–relevance in research [27, 33]; focusing on emerging phenomena [34, 35]; positioning research implications to the higher business level rather than the narrow level of the marketing department [36]; running role-relevant research driven by a deep understanding of the core tasks of the marketing department; translating research results into actionable recommendations [23, 37].

In sum, marketing research has been criticised for not having an impact on practice since it is primarily focused on writing for other scholars and not for practitioners who might benefit from marketing research to address practical issues. Equivalently, publishing marketing research that is more useful for practitioners implies that there should be a well-functioning nexus between the theory and the marketing tools and techniques that practitioners need to deal with practical issues. In the context of social media, we still don't know whether academic publications have an impact on practitioners. In other terms, one may ask whether social media practitioners read academic articles or refer to these publications in their practices.

Our paper considered the gaps between the theory and practice of social media and identifies where they exist. Some possible explanations for the gaps can be explored which may be of interest to both academics working in the field.

## Study 1

We conducted a bibliometric study, consisting of the collection, summarizing, assessing, and monitoring of published research, to create an up-to-date overview of the current marketing research on social media and statistically assess the associated literature [38, 39]. Scopus, one of the most complete databases of academic articles, served as our data source. It indexes 12,850 periodicals in various categories and contains articles published since 1966. Scopus was chosen over Web of Science for two reasons. First, as researchers faced a trade-off between data coverage and cleanliness, Scopus has been discovered to have a larger coverage (60% larger) than Web of Science [40]. Second, bibliometric studies in marketing research often

employ only one database to avoid data homogeneity problems that could arise when using numerous databases [41].

To search the database, we first identified two keywords related to our study: "social media", and "social media marketing", we ran a query using a combination of these keywords (adopting the Boolean operator "OR") in the fields related to "title," "abstract," and "keywords." We considered works published only in business journals until October 2022. Proceedings, book chapters, and books were excluded from further consideration. To filter this data, we relied on a screening process. Documents were excluded based on whether they are not published in English and/or not available for the project team.

We placed the highest priority on maintaining the reliability of our dataset, which we accomplished by adhering to protocol. This protocol was carried out in four distinct phases: identification, screening, eligibility, and inclusion, as elaborated in Fig 1. Using Mendeley's robust features, we structured all identified studies in an organized format consisting of author names, titles, and publication years. Additionally, we conducted a thorough check to detect and remove any duplicate studies, ensuring the dataset's cleanliness and integrity.

After applying our selection criteria rigorously, our initial search of the Scopus database produced a substantial dataset comprising 5345 research works. This dataset encompassed a wide range of information, including author names, article titles, the countries of corresponding authors, publication counts, comprehensive citation statistics (total citations, average article citations, and the number of citing articles, both with and without self-citations), journal sources, keywords, geographical distribution by countries, and author-level metrics. A detailed workflow outlining our selection process is depicted in Fig 1, providing a comprehensive overview of our systematic approach.

To further enhance the comprehensiveness of our research, we implemented a backward search strategy. In this phase, we scrutinized the reference lists of the retained studies for our final review but did not identify any additional studies relevant to our research objectives.

Once we finalized our ultimate dataset, we examined the complete text of each article. We extracted and organized all pertinent information essential for our review. To streamline this process, we developed a structured data extraction tool specifically designed to record and concisely summarize the crucial details necessary to address our research inquiries. This approach aimed to minimize potential human errors and enhance procedural transparency.

The data coding phase unfolded in two distinct steps. Initially, we subjected the data extraction form to a rigorous pilot evaluation using a select sample of the finalized articles. Two of our co-authors independently conducted data extractions from this sample, allowing for a meticulous cross-check to identify and rectify any technical issues, including completeness and the form's usability. In the second step of data coding, each article received a unique identifier. One co-author examined the complete text of each article, coding the data into specific categories, such as article title, publication year, geographic market focus, and research theme. To ensure the utmost accuracy and reliability, a second co-author rigorously reviewed the extraction form and conducted a random sample check for cross-validation. Any discrepancies or disagreements that arose during this process were thoroughly discussed and resolved to maintain the integrity of our data coding efforts.

Then, we proceeded to conduct a comprehensive performance analysis. Within this evaluation, we scrutinized various metrics to gauge the scholarly contributions. Among the myriad measures assessed, two stood out as particularly prominent indicators of research impact. The first criterion was the number of publications produced per year or per research constituent, serving as a robust proxy for productivity and output. The second metric revolved around citations, a paramount gauge of the work's influence and impact within the academic community.

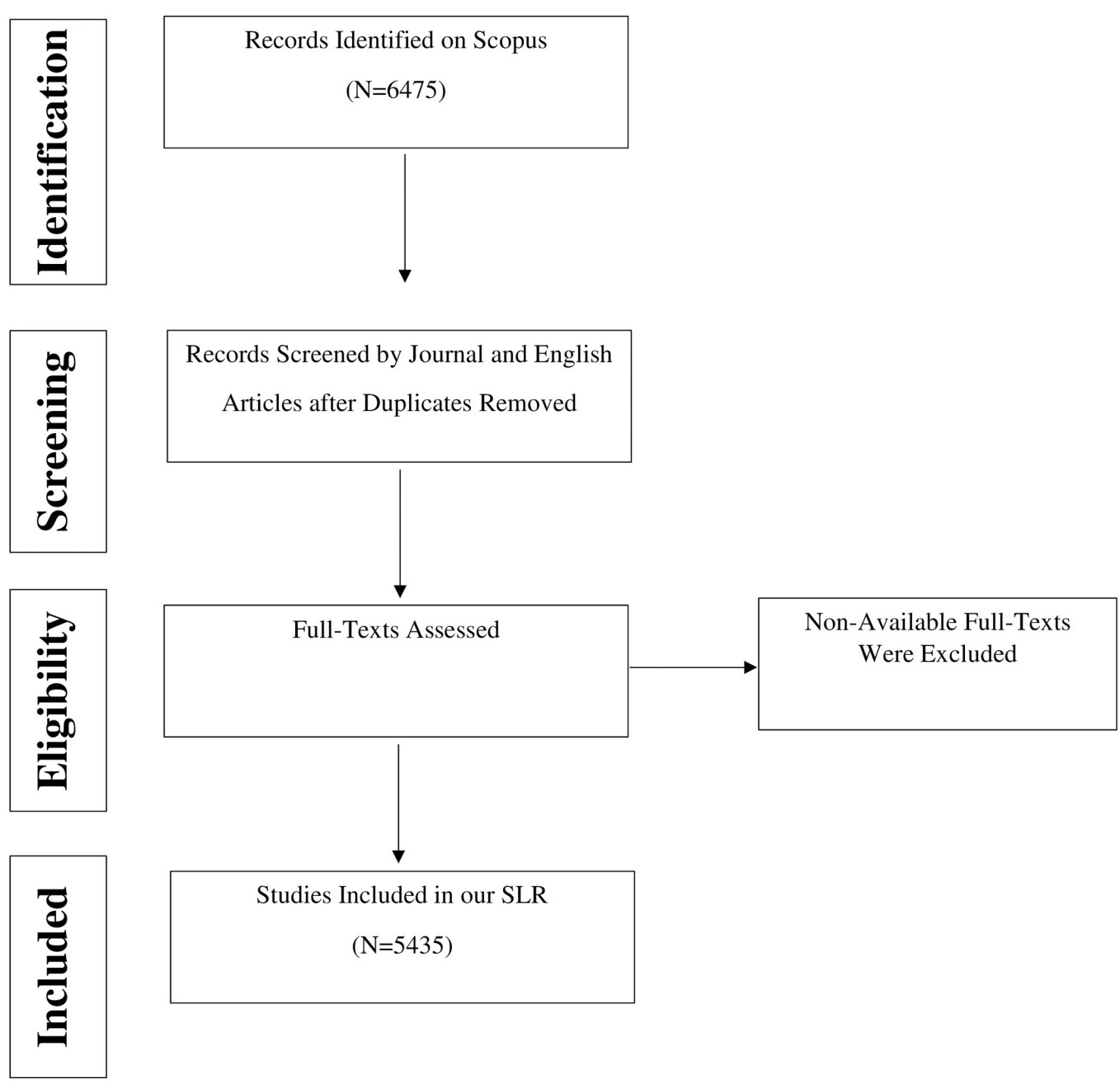

**Fig 1. Study 1 protocol.**

As mentioned by Donthu et al. [42], these dual facets—publication and citation—equally underscore the multifaceted nature of scholarly contributions.

Fig 2 shows the evolution of publications on social media research. Academic research was keeping up with the growth of social media platforms. In fact, since 2010, the number of social media users has significantly increased, the number of social media networks has grown, and the social media platforms evolved from direct electronic information exchange to virtual gathering place.

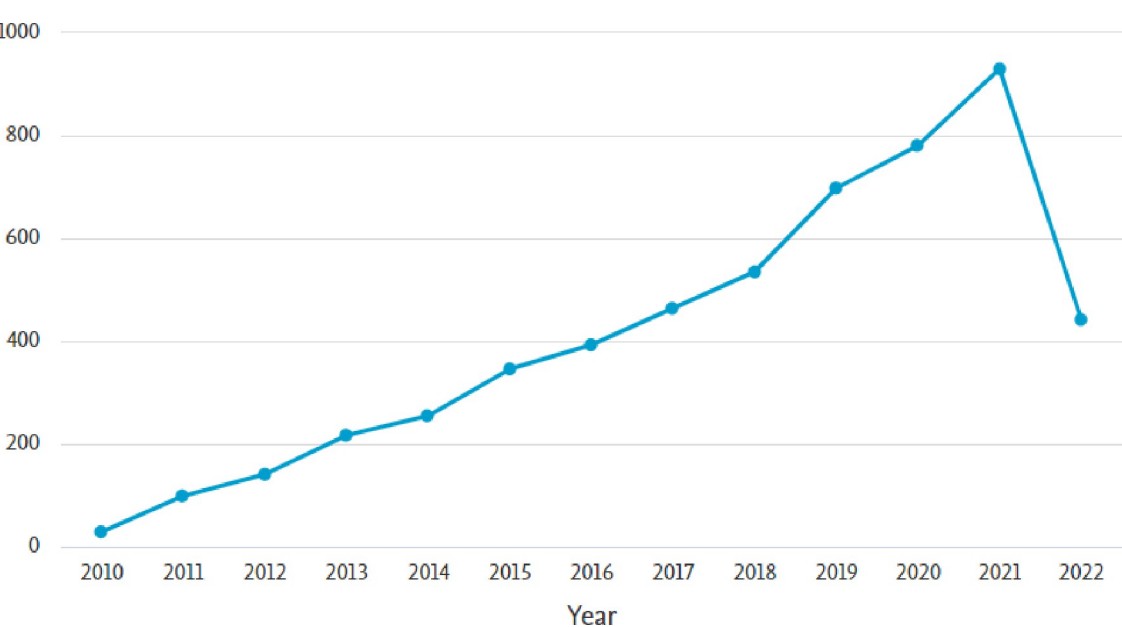

**Fig 2. The evolution of social media research.**

These publications have collectively accumulated a total of 2,351 citations. Further dissecting this data, we found that the average citation per document stands at 0.43. Moreover, the h-index, a key indicator gauging both the productivity and influence of a researcher's body of work, stands at 16. This signifies that a minimum of 16 citations have been garnered by these publications.

Overall, the studies have been conducted in 52 countries. The greatest share of research has been carried out in the United States, followed by the United Kingdom, Australia, and China. Business journals publishing on social media were also listed in our findings. They included a range of marketing journals (e.g., Journal of the Academy of Marketing Science; International Journal of Research in Marketing; Journal of Marketing; Marketing Intelligence and Planning; Journal of Interactive Marketing; Journal of Research in Interactive Marketing; Psychology and Marketing), business research journals (Journal of Business Research; EuroMed Journal of Business; Business Horizons; Journal of Business Media studies), and management journals (e.g., European Management Review). In general, the Marketing journals and business research journals dominated this list. The most cited paper was the classical article on the challenges and opportunities of social media published by Kaplan and Haeinlein in Business Horizons, which has received more than 27000 citations. Next, we found three papers with more than 5000 citations and 10 additional ones with over 2000 citations.

In our study, the abstracts were assessed for the keywords "theory" and "model" to identify theories and models that the social media study added to, and to explore the many theoretical lenses used to guide the research. In total, 321 papers had an abstract that included a theory or model. There were 19 different theories initiated from those works. Theory of gratification, the theory of technology acceptance, and the theory of planned behavior are the most used theories in social media research.

Next, we conducted a keyword co-occurrence analysis aimed at identifying the main keywords i.e., academic research on social media. By investigating the relationships between keywords, keyword co-occurrence analyses helped us to represent and comprehend the literature of a scientific topic. VOSviewer package of Van Eck and Waltman [43] was used to generate

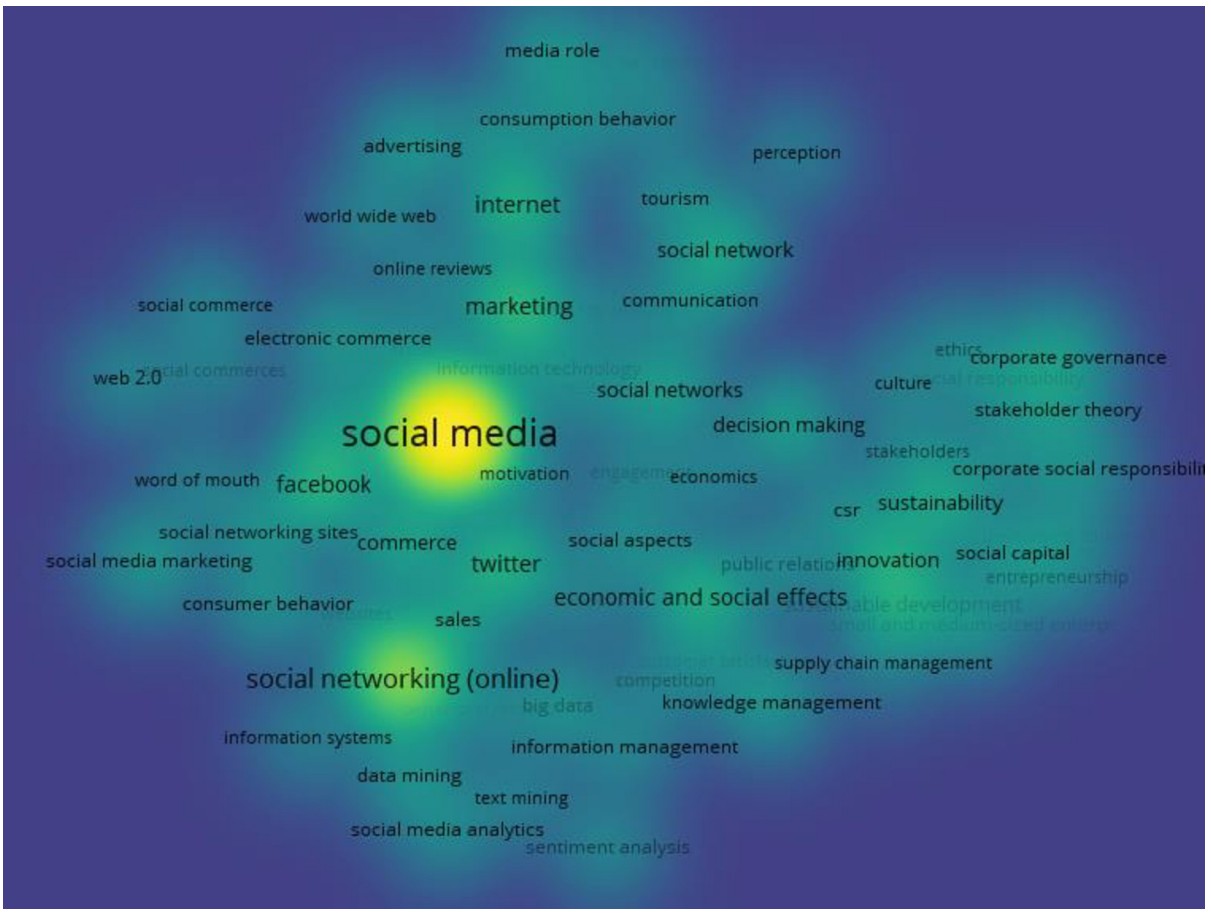

**Fig 3. The main keywords in social media research.**

bibliometric analysis which has been widely adopted in the literature [44]. VOS is superior to multidimensional scaling for constructing bibliometric analyses and maps [43], so we did not involve multidimensional scaling. The results of our keyword analysis are shown in Fig 3. The size of the circles in the graphic representation indicates which keywords had the highest levels of co-occurrence throughout the examination of keyword co-occurrence. Social networking platforms, consume behaviour, marketing, sustainability, and economic and social effects are among the most important keywords used by social media researchers.

Social media research has identified several prominent topics that have garnered significant attention in recent years. One of these is the impact of social media on mental health [45]. Researchers have been exploring how various aspects of social media use, such as the quantity and quality of online interactions, content consumption, and platform design, influence individuals' mental well-being. This includes investigations into the links between social media use and consumer well-being [46].

Another critical area of focus in social media research is the spread of misinformation and fake news [47]. The mechanisms behind the dissemination of false information on social media platforms have been extensively examined, along with their effects on public opinion, trust, and political polarization. Researchers have sought to understand how the algorithms, echo chambers, and filter bubbles on these platforms contribute to the propagation of misinformation [47].

Influencer marketing is another hot topic in social media research [48]. The effectiveness of influencer partnerships, issues related to authenticity, and the ethical considerations surrounding sponsored content have all come under scrutiny. Researchers are also exploring how influencers shape consumer behavior, affecting choices and preferences [49].

Political communication on social media has attracted significant attention, particularly regarding its impact on election campaigns, policy-making, and public discourse. Scholars have examined how algorithms, the presence of filter bubbles, and the formation of echo chambers can influence political opinions and contribute to polarization [50].

Privacy and data security issues are ongoing concerns in the realm of social media research. These studies investigate user privacy, data breaches, and the effects of privacy settings on various social media platforms [51].

Research into user behavior and engagement on social media platforms is fundamental for understanding trends, virality, and the factors that drive user interactions. Numerous research studies have delved into the impact of social media on consumer behavior, offering valuable insights into this dynamic relationship [52]. The concept of social proof, where people tend to follow the actions and preferences of others, is well-established in social media research [53]. The fear of missing out (FOMO) also drives consumer behavior, as limited time offers and exclusive deals on social media can prompt quick purchasing decisions [54].

Several emerging trends in social media research are shaping the field's future direction. One of these is the integration of artificial intelligence (AI) and machine learning into research methodologies [55]. AI is being used for sentiment analysis, content recommendation, and identifying trends within large datasets [56].

Blockchain technology is also gaining traction as a means to enhance trust and transparency in social media interactions. Researchers are exploring its potential in content authentication and combating fake news [57].

Ethical considerations surrounding AI algorithms on social media platforms are a growing concern. Research in this area focuses on issues of bias, fairness, and the ethical responsibility of tech companies in algorithm design and implementation [58].

Cross-cultural and global perspectives in social media research are becoming more prevalent, with studies investigating how social media usage varies across cultures and regions and the global impact of social media trends [59–61].

Looking ahead, future research in social media could consider the long-term effects of social media usage on individuals and societies, including potential generational attitudes and behaviors. Advocacy for increased algorithmic transparency on social media platforms and the study of its impact on user experiences and content distribution is another important direction for future research. Research into how humans and AI can collaborate to enhance content moderation, fact-checking, and information verification on social media platforms will become increasingly relevant. Finally, as concerns about environmental sustainability grow, future research could investigate the environmental impact of data centres and the carbon footprint associated with social media platforms.

### Study 2

Building upon the insights garnered from our initial study, our research journey continued with two subsequent investigations. The second study delved into the world of social media practitioners, extracting valuable feedback and information. In parallel, our third study synthesized a comprehensive list of suggestions tailored specifically for academic researchers. These two complementary endeavors, while distinct in focus, formed integral components of our overarching quest to bridge the gap between theory and practice in the realm of social media.

Numerous academics and commentators have recently claimed that marketing scholarship has stopped being sufficiently creative and has grown more disassociated from actual practice. A rush of recent special journal issues, editors' forums, and studies on the seeming research/ practice gap in marketing have been linked to such worries. While some contend that the goal of marketing research should be to enhance rather than merely describe, understand, or criticise marketing activity, a possible divide between practitioners' and academics' concerns appears to have formed.

Academic research, characterized by its systematic inquiry, rigor, and peer-reviewed dissemination, represents a cornerstone of knowledge production and dissemination across various fields and disciplines. Yet, the extent to which social media practitioners engage with, trust, and value academic research remains a subject of limited empirical inquiry. This gap in our understanding is particularly pertinent given the increasingly complex and intertwined relationship between social media and academia.

While a growing body of literature has explored the impact of social media on academic research dissemination and public engagement [62, 63], relatively few studies have focused on the reverse perspective—how social media practitioners perceive and interact with academic research. In response to this gap, Study 2 aims to delve deeper into this important facet of the digital age information ecosystem.

The sample frame for this study consisted of 280 social media practitioners. Emails were sent to 441 marketers explaining the project and posing their participation. To improve response rates, a cover letter and a survey instruction letter were sent to all potential respondents [64]. The overall response rate from the participating companies was 63.5 percent. Table 1 displays the characteristics of the participants. To facilitate the process of reaching social media professionals, we collaborated with a local professional marketing body. This collaboration provided us with access to their membership database and allowed us to leverage their network to identify and contact potential participants. Our data collection initiative unfolded over the course of three months, spanning from November 2022 to January 2023.

The selection of a sample frame comprising 280 social media practitioners for this study was well-justified on several grounds. Firstly, this choice was rooted in the research's primary objective, which seeks to gain valuable insights into the perspectives of professionals actively engaged in the field of social media. Moreover, the sample size of 280 was both practical and feasible, considering the available resources and the capacity for efficient data collection and analysis. This size also ensured the statistical significance of the study's findings, reducing the likelihood that results are simply due to random chance. Ultimately, this choice of sample frame aligned with the study's research goals, methodological considerations, and ethical principles, strengthening the validity and reliability of the study's outcomes.

Results showed that only 2% read an academic paper every quarter. Less than 1% of our respondents believed that academic research had an impact on their decisions and activities. Participants of our survey indicated that social media platforms, practical eBooks, specialised websites, and newspapers and magazines are their main sources of knowledge. Fig 4 shows the responses of the participants to the following question: What media or tools do you use to acquire knowledge in social media?

Participants mentioned that they use the previously mentioned tools to mainly gain more information about the latest trends in the fields of social media and technology. Others read market analysis reports and practical reports, while the rest use the templates developed by marketing institutes and digital marketing agencies (See Table 2).

Respondents shared their perceptions of marketing academics. Academics were regarded as being elitist because they use their own jargon, speak in convoluted scientific terms, strive to publish at all costs, and do little to advance practice. Moreover, most of our participants raised

**Table 1. The characteristics of the respondents.**

| Years of Experience | |
|---|---|
| <5 | 15.3% (43) |
| 5–9 | 40% (112) |
| 10–19 | 34% (95) |
| >20 | 10.7% (30) |
| Education | |
| High School | 0.7% (2) |
| Bachelor's | 78% (219) |
| Master's | 21.3% (59) |
| Degree | |
| Business | 82% (229) |
| Graphic Design | 7% (19) |
| Computer Science | 11% (32) |
| Job Title | |
| Social Media Specialist | 27.5% (77) |
| Social Media Manager | 7.5% (21) |
| Social Media Director | 11.8% (33) |
| Social Media Coordinator | 22.5% (63) |
| Social Media Consultant | 6% (17) |
| Social Media Associate | 10% (28) |
| Social Media Executive | 14.7% (41) |
| Country | |
| France | 20% (56) |
| U.S.A | 19% (53) |
| UK | 11% (31) |
| Netherlands | 7% (19) |
| Singapore | 11.4% (32) |
| Australia | 14% (39) |
| Lebanon | 3% (9) |
| UAE | 14.6% (41) |

negative criticisms towards the academic research on social media. For them, academic research lacks usefulness, relevance, and visibility (see Table 3). Respondents were also asked what the ideal focus of an academic journal should be on. Results show that the most important areas of focus are: (1) practical cases, (2) best practice sharing and (3) dissemination of new ideas. According to the respondents, the least focus should be placed on theoretical models.

## Study 3

Building upon the insights gained in our second study, we transition into our third investigation, which, like the second, continues to focus on social media practitioners. This continuity underscores the importance of deepening our understanding of their experiences, challenges, and expertise within the ever-evolving social media landscape. In this manner, our research endeavors maintain a cohesive narrative, as we gather comprehensive feedback from practitioners to inform our ongoing quest for practical solutions and academic contributions.

11 online focus groups were held with 69 social media practitioners between March and June 2023. Focus groups are a popular qualitative research method for producing cutting-edge

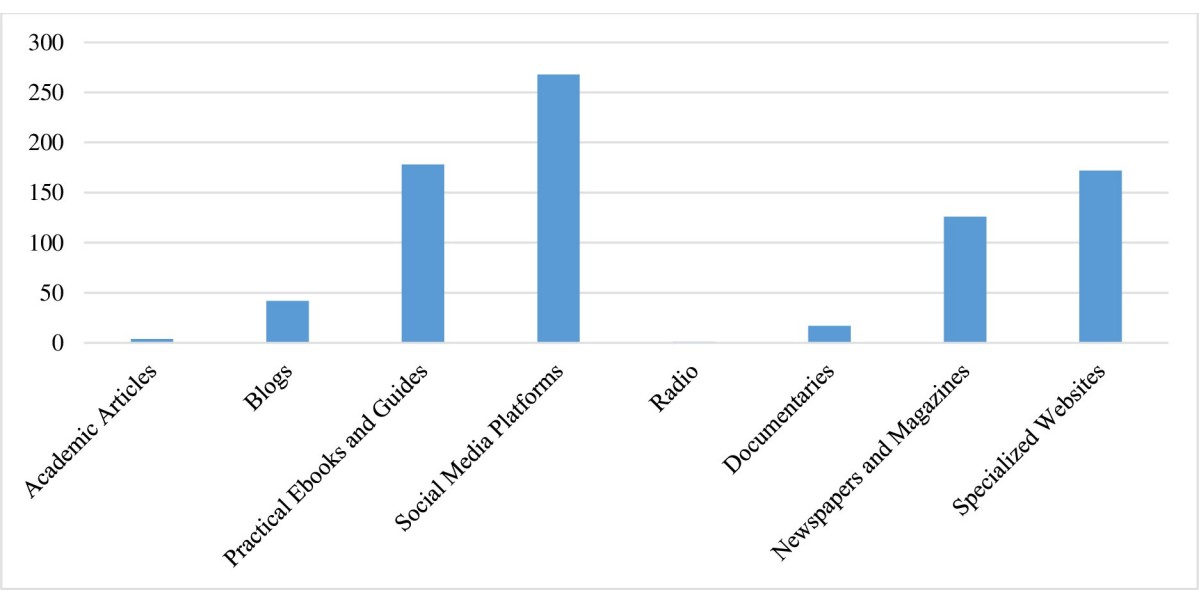

**Fig 4. List of media tools that practitioners use to acquire knowledge.**

findings. This approach was designed and justified to ensure the rigor and comprehensiveness of our research. By opting for a series of focus groups, we aimed to capture a diverse range of perspectives within the field of social media, considering its dynamic nature and the variety of roles practitioners may hold. Smaller groups can lead to more in-depth discussions and allow each participant to contribute meaningfully [65]. Furthermore, conducting these sessions during a specific time frame ensured that our data remained relevant and reflective of contemporary practices. The approach also provided opportunities for data validation and triangulation, enhancing the trustworthiness of our findings. Overall, the methodology aligned with the study's objectives, maximizes data richness, and accounts for practical constraints, ultimately enhancing the robustness and credibility of our research.

The moderator and participants must be chosen for the online focus group to grow well. For the role of moderator, the focus group was facilitated by one of the researchers with experience in conducting online focus groups, whose main role was to encourage an open and relaxed discussion, keep the discussion relevant and probe into areas that needed clarification. A relaxed atmosphere was created to improve interaction and the free flow of ideas and opinions. 'Starter' questions were used at the focus group sessions to seed the discussions about the role of academic research.

We did not predetermine the number of focus groups, but we did adhere to the principles of saturation and ceased gathering data when there was no more material to add. The focus groups lasted between 62 and 91 min, with an average length of 73 min. We purposively selected information-rich participants [66] by using the authors' networks and snowball sampling. All the participants had a strong understanding of social media, having more than three

**Table 2. What do practitioners read?.**

| | |
|---|---|
| Market Analysis | 20% (56) |
| Technology and Social Media Trends | 63.2% (177) |
| Professional/Practical Reports | 11.5% (32) |
| Templates | 5.3% (15) |

**Table 3. Respondents' views on academic research on social media.**

| | |
|---|---|
| Lacks Relevance to the Challenges of Professional Practice | 73% |
| Not Interesting | 69% |
| Not Visible for Me- "I have not read or seen an academic paper before" | 67% |
| Useless | 63% |
| Important- "academic articles would help them to do the job better" | 6% |

years of experience in digital marketing. An emailing list of marketing professionals was already developed by the researchers. Those professionals already participated in a previous survey or engaged in an academic activity (example: guest talk, judging a case competition, industry speech to the students). The sample frame for this study consisted of 69 marketing professionals in Singapore, France, and Lebanon. Emails were sent to 130 marketing professionals explaining the project and posing their participation. All potential respondents received a cover letter and a letter outlining the survey's requirements to increase response rates. The overall response rate from the participating companies was 53 percent. Industries included high-tech, investment, banking, media, logistics, retailing, and healthcare. 55.07% of the sample salespeople were males with a mean age of 33 years.

Each focus group was transcribed and reviewed by an independent researcher. Using computer-based qualitative analysis software, the authors created the initial codes (QSR NVivo 12 Plus). To work methodically through the complete focus group data set and give each data item its full and equal attention, the authors scheduled frequent follow-up meetings. Then, the authors used a data-driven thematic analysis approach to look for themes that were strongly connected with the data but unrelated to the topics posed during the focus groups. To ensure that the themes formed a logical structure, the authors revised and improved the topics. To make sure that each theme appropriately reflected the meanings visible in the entire data set, the authors specifically went back to the transcripts. The final thematic categories underwent numerous rounds of editing to verify that they accurately translated the empirical data and were free of overlapping meanings. They also established connections between the detected themes and previously published works of literature. The trustworthiness of the authors' later findings, which were demonstrated and backed by a comprehensive collection of data quotations, required such iterative analysis to be improved. Finally, the authors created a report by summarizing the results and highlighting how the new research has added to the body of knowledge. Reports were sent to all the study participants to obtain their feedback. The feedback validated the findings.

Before conducting a thematic analysis, textual data were analyzed using VOSviewer software. The software can automatically and precisely identify the keywords that frequently exist in a large text. VOSviewer analysis needs data files in.txt or.csv format for analysis. As such, all data files were converted from.docx to.txt format and uploaded to the software. The research generated many keywords, each of which was represented by a circle based on how frequently it appeared in the transcripts. The higher the frequency of occurrence of a theme in the textual data or transcripts, the larger the size of the circle (See Fig 5). We identified 10 topical keywords: 1) relevance; 2) automation; 3) artificial intelligence; 4) technology; 5) exposure; 6) partnership; 7) big data; 8) innovation; 9) analytics; and 10) trends. These keywords are intelligible with the importance of publishing relevant and visible academic research to practitioners. This analysis also shows that technology, automation, and artificial intelligence are among the main interests of our participants.

In this section, we present the findings of our thematic analysis. To answer our research questions, our findings shed the light on 1) the problems of academic research on social

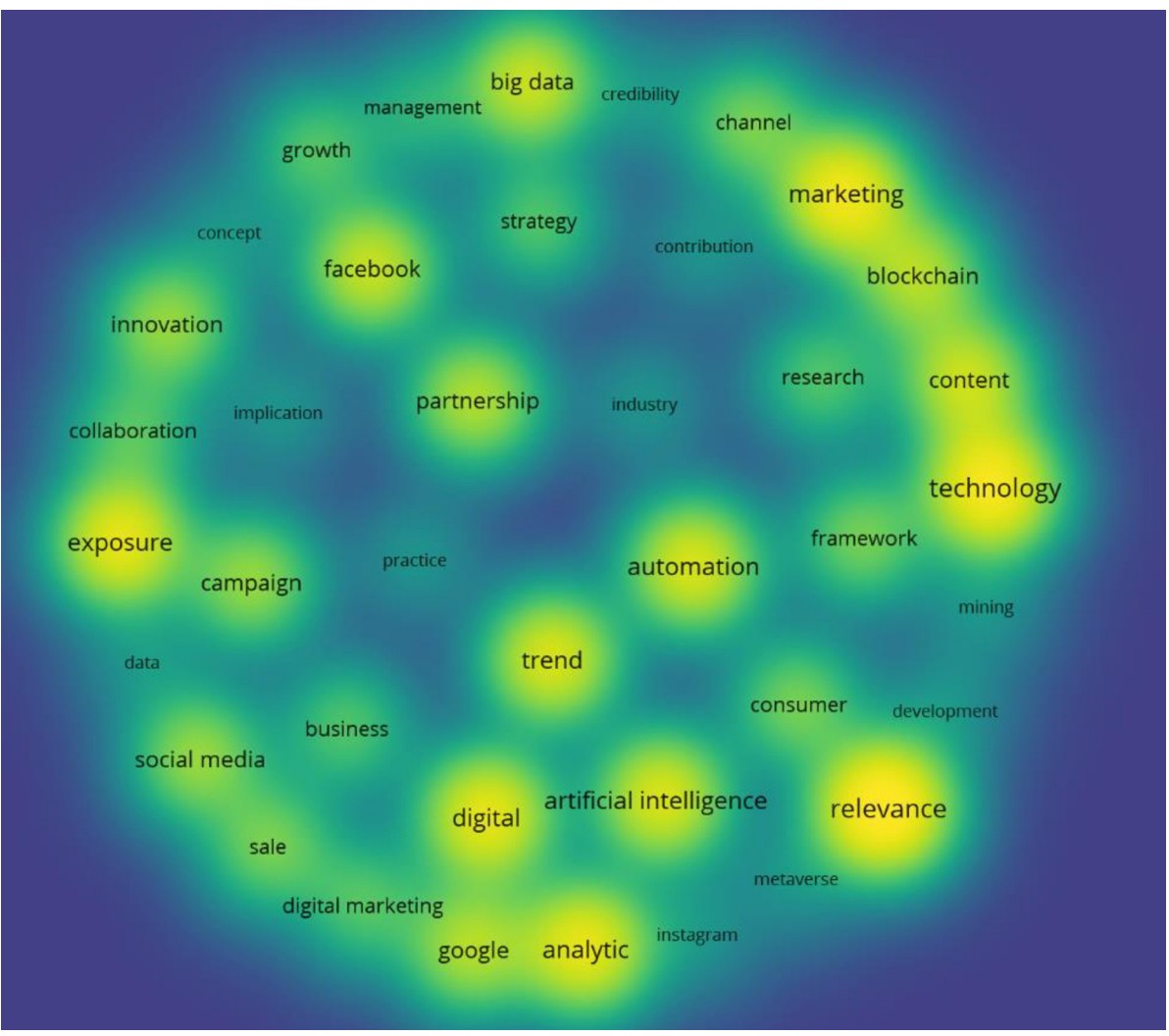

**Fig 5. The main themes.**

media; and (2) the participants' recommendations to improve the managerial and practical impact of academic research on social media.

A primary finding of this research was related to the current challenges the academic research on social media. All our participants thought that the lack of visibility of academic articles is the main problem of academic research. All of them mentioned that they are not aware of the academic journals in the field of social media and are not exposed to the academic articles. Participants believed that academic institutions and researchers don't promote research works to practitioners, as described in the following paragraph:

> *I think academic research is not visible for professionals like us [. . .] I believe that academics need to invest more time with practitioners and present the results of their research. As things stand like this, we are not even aware of what researchers are doing.*

(Participant 13, Digital Marketing Manager, 10 years of experience).

The participants also mentioned that the lack of relevance for research in social media—is overwhelming: the gap between social media research and its practice looks abysmal, even irreconcilable, and practitioners' interest in the research produced by researchers is very limited. The relevance dimension relates to the originality of the results. However, according to the participants, the nature of originality differs between researchers and social media managers. The originality expected by social media practitioners is established in reference to their daily practice (there is originality when the results question the practices in use) whereas the originality for researchers has as reference the universe of theories (there is originality when the results question existing knowledge). And this has implications for the researcher concerned with relevance. If he or she evolves in the world of theories and is, therefore, able to detect what constitutes original results, as quotes of participant 24 depict:

*We are looking for new and original contributions [. . .] We expect to read something that we don't know and can help us in our job.*

(Participant 24, Social Media Coordinator, 5 years of experience).

The participants felt that there is an obvious time gap between researchers and practitioners: Thus, the researcher took liberties with research time which, in the eyes of the marketers, sometimes made the results obsolete. This perspective of a two-speed world (always slow for research, very fast for marketers) brought with it the idea that researchers are disconnected from business constraints and realities, as depicted in the following quote:

*We don't work much with marketing researchers because there is too much of a time* gap.

(Participant 43, Marketing Manager, 13 years of experience).

Another problem noticed by our participants concerns editorial style and the dissemination of the writings. On the one hand, practitioners knew that vocabulary is an element of a researcher's scientific capital. It is a proof of legitimacy and a mark of identity. On the other hand, they also considered the scientific writing style complex and difficult. In fact, according to our participants, the latter do not read academic articles because they are not trained to understand their content, which is often too specific, abstract, written in specialised jargon, and peppered with references and methodological details. Social media practitioners look for easy-to-understand texts that address the real world and offer cutting-edge ideas, digital marketing trends, or advice that help prevent or solve crises, address challenges, or stay competitive in the digital world.

For our participants, the questions addressed in the research work must be in phase with the concerns of social media practitioners and the challenges facing companies. They must therefore deal with current, and even future, topics, as these are the ones that interest practitioners and can help digital marketers make a difference, stand out and progress in their practices. Therefore, many of our participants said that the topics that social media researchers tackle may neither be important nor interesting for them. In this sense, participant 3 expressed the following:

*Research is not at all intriguing. It is very difficult for me to understand and covers subjects that I don't find interesting.*

(Participant 3, Social Media Specialist, 5 years of experience)

The participants listed several recommendations to improve the practical impact of academic research on social media. First, there is a need to improve the visibility and exposure of

the academic articles to the practitioners. Academics can think about organizing conferences and seminars to present their results to practitioners. Joint seminars involving academics and practitioners might be planned in this order to increase the practical value of academic research. Also, academics, according to our participants, can publish the results of their results in practice websites. Researchers should also be more active on social networks and try to use less scientific and more practical language in their blogs to explain the importance of their results to social media practitioners. Participants think that researchers should share specific recommendations for practitioners through YouTube. For instance, they can practically explain how to increase social media engagement and how to improve the online reputation of a company. Moreover, professional associations and professional institutes can play an important role in transmitting academic research findings to practitioners.

In addition to presenting at practitioner conferences, writing in traditional crossover journals for practitioners as well as in shorter pieces like op-eds and blogs, and attracting the attention of those who publish columns, blogs, and articles about research for practitioners, participants discussed other ways that scholars can share pertinent research insights with practitioners. Second, participants suggested involving practitioners in research, conducting research in partnership with companies. Implementing collaborative research is not easy, however. It assumes that researchers and social media managers can navigate both worlds and comply with the rules of the game specific to each universe. Participants emphasised the importance of developing ecosystems that bring together researchers and companies to pool resources from both sides to tackle shared issues. Offering a place to share problems and solutions is relevant and improves the efficiency of efforts, whether through knowledge sharing, financial support mechanisms (sponsorship), or—the implementation of joint field studies. Social media practitioners should play an important part in the co-production of research and so shape how new social media plans and ideas are developed. Participant 19 said:

*I think researchers need to bridge the gap with professionals [. . .] A key element here is: collaboration. The collaboration will help both parties. The researchers will be able to co-create knowledge with the practitioners, and the latter will get new perspectives from the researchers.*

(Participant 19, Marketing Executive, 20 years of experience)

Additionally, social media practitioners can collaborate with researchers to co-produce research, as shown by examples where they have served as data sources, recipients, commissioners, endorsers, and co-researchers on collaborative projects. For two crucial reasons, it is uncommon for practitioners to write on discipline-based research. The first is that academic journals have historically been less likely to publish articles on topics that are of interest to both academics and practitioners and on which they may collaborate in research. If any such research is done, it might be published in journals geared toward practitioners, but these publications don't seem to get the attention or credit required to improve an academic's reputation or career. Second, because practitioners lack training in the vocabulary, tools, and processes of research analysis, co-production is uncommon. To promote the generation and dissemination of information, business schools might suggest research training for practitioners and can create networks between practitioners and academics.

Third, practitioners recommended research to tackle specific, relevant, and "real-world" topics. Practitioners look for research that can provide new insight and information, which goes beyond intuition. This relevance non-obviousness test determines whether research meets or exceeds a practitioner's intuition. So, the research topic must evoke a real marketing problem and the focus should be directed toward helping practitioners with new insights or knowledge. Then, researchers should convert this info into practice and communicate the

**Table 4. List of suggested topics.**

| |
|---|
| Benchmarking between digital marketing automation tools: which one is the most efficient? user-friendly? effective? |
| Innovation of new practical social media frameworks |
| Development of social listening frameworks |
| Studying the integration of Augmented Reality in social media |
| Benchmarking the ROIs of social media marketing campaigns between channels |
| New frameworks to incentivise consumers to buy directly from social media channels |
| Innovation of social audio frameworks |
| Studying the effects of social media on consumers |

same in jargon-free language. Social media research should concentrate on know-how rather than know-what and need to mainly rely less on theory to the virtual exclusion of practitioner utility. It is also suggested that researchers consider the results of surveys that some academic or professional organizations periodically conduct to identify practitioner needs and challenges.

Making social media research more relevant does not mean neglecting its rigor, but rather reconciling these two qualities. This "research/teaching/transfer" synergy implies changes in thinking, acting, and managing on the part of all stakeholders: faculty, policymakers, and the management of educational institutions. Participants proposed recommendations for promoting relevance at each stage of this process. Thus, according to them, research must: 1) focus on marketing problems that face practitioners and on variables that practitioners can influence; 2) analyze the interactions between, people, organizations, and the industry and their impact on marketing performance indicators deemed important by practitioners; 3) generate results that cannot be deduced intuitively and that are quickly transferred to practitioners.

Furthermore, participants thought that there is a need to investigate the effect of media on consumers, value formation and their interplay with purchase decisions, and optimization of social media marketing tools that marketing practitioners can use in their everyday jobs. Table 4 presents examples of suggested topics by our participants.

Fourth, social media research needs to provide recommendations that motivate practitioners to act on the results. Therefore, when possible, researchers should use examples to illustrate how their findings should affect social media practitioners' actions while also defining the context in which those findings are most likely to be applicable. This entails including—and treating seriously—a part in papers titled "implications for practice" and, ideally, having relevant practitioners review versions of this section. In the eyes of practitioners, it appears necessary for researchers to adopt a forward-looking posture and look to the future rather than simply explaining what has happened or is happening today, as described by participant 58:

*I expect recommendations from the part of researchers on how to prepare for the future of digital marketing.*

(Participant 58, Digital Marketing Manager, 15 years of experience)

Fifth, participants called for new thinking on research analysis methods:

*We, the marketing directors, need models to help us make decisions and make judgments, and this is a strong expectation from researchers.*

(Participant 40, Social Media Specialist, 7 years of experience)

As far as data mining and retargeting are concerned, social media practitioners are waiting for models to better manage data flows. Quantitative approaches are therefore not excluded, but they must be combined with interpretative approaches, backed by psychology, ethnography, or anthropology, to better understand and anticipate consumers' choices and decisions.

Sixth, the skills needed to thrive in the field of digital marketing are numerous and diverse because it mixes traditional marketing, web design, SEO, analytics, content management, and much more. Tech-savvy digital marketers are always trying to learn more. For instance, digital marketers need to be familiar with data analytics. To make wise decisions, it is advised to regularly update the database and remove irrelevant information. Additionally, having a basic understanding of HTML and CSS can help digital marketers if they will use WordPress. So, there is an emergence in the required technical knowledge and skills. Our participants advocated for more technical academic research where marketing researchers collaborate with researchers from computer science and data analytics backgrounds. Such research would contribute to the multidisciplinary character and hybrid task structure of the digital marketing profession, which uses technology, analytics, and marketing.

Finally, participants indicated five areas where researchers can help digital marketers to be more knowledgeable and acquainted: 1) the use of big data in the social media field; 2) the AI-powered technology in digital marketing; 3) the characteristics of emerging markets; 4) the features of metaverse marketing; and 5) the use of blockchain technology in digital marketing and social media.

## Discussion

The study has several key implications for future social media investigations. First, a direct correlation exists between academic research and the evolution of social media in terms of users, platforms, and medium. Nevertheless, there is a gap between social media studies and their influence on the marketing function. The problem can be attributed to laxity on the part of digital marketing institutes and centers linking practitioners and researchers [67]. The onus is on digital marketing institutes to assume the intermediary role between academic researchers and social marketing professionals.

Today, much social media research is invisible to practitioners because the system is primarily self-referential. Authors are forced to decide whether to undertake socially impactful research or write publications that are only academically impactful [68]. The various participants in the focus groups provide potentially practical solutions to the issue. A close relationship between academic researchers and practitioners increases the chances of implementation. Collaborations help researchers play the role of participant-observer in practitioners' responsibilities and decision-making. Therefore, social media investigations can consider academic and informal aspects to enhance readability and generate a significant audience.

Another reason that could explain the gap between practitioners and researchers is that the former focus on publishing instead of engaging their target audience. Thus, a disparity exists between the expectations of the marketing professionals trying to transform their work for industry consumption and the researchers' interests. Additionally, social media marketers work in an exciting field, but academic researchers continue to ignore this vital fact [69]. Cadotte et al. [70] support the perspective and contend that practitioners feel that academics consider themselves elitists who speak their jargon and write in complex scientific language. The issue explains why 69% of our respondents stated that they do not find academic articles interesting. Although theoretical models are critical in research studies, authors should focus more on practical cases, best practice sharing, and disseminating new ideas. The fast-paced and dynamic nature of social media marketing [71] requires companies and marketers to rely

on scientific studies to make informed decisions [69]. However, academic journals are characterised by the slowness of their review process. Managerial fads are often gone by the time the articles that dissect them are published.

Our research also focuses on academic research challenges regarding social media and implementing participants' recommendations to enhance practicality. The lack of originality in academic papers means that marketing practitioners do not find the articles relevant to their practice. According to Roberts et al. [19], professionals want researchers to publish studies that value practical relevance, but academics favour journals with a high impact factor. Young and Freytag [72] indicate that the only approach to bridge the gap between research studies and their influence function is through successful collaborations between academic researchers and professional practitioners.

## Conclusion

The findings of this paper show that academic research on social media is growing in terms of the number of publications but is struggling in three areas: visibility, relevance, and influence on practitioners. Our findings outlined some difficulties in bridging academic research and social media usage. We offered some recommendations for improving the interaction between research and practice after focusing on the viewpoints of the practitioners. Thus, we discussed why research relevance is important and how scholars might raise the relevance of their research in an effort to inspire academics to produce research that is more pertinent to social media practitioners.

Our findings should be of interest to marketing researchers and academic institutions. We presented a detailed snapshot of guidelines to publish more impactful research works. Collaboration between scholars and practitioners is an important area where the gap could be closed. Collaboration with professional organisations and businesses should already be under progress. In order to advance toward positive developments in the interaction between the academy and practice, collaboration will result in stronger links between social media research and the instruments used in professional practice. Another position consists in asserting that the lack of relevance of the knowledge produced by social media research is the direct consequence of its mode of production. By raising to the rank of dogma a paradigm inherited from the hard sciences, the scientific community has, so to speak, dried up the marketing discipline. As a result, the findings of the social media research conducted by the researchers have lost interest. To make social media research more relevant, it would be necessary to change the way research is conducted. An initiative such as the creation of the d.school at Stanford University, a program that fundamentally changes the way of thinking about teaching and research based on the design sciences paradigm, is part of this conception, but other approaches such as critical digital marketing studies, critical realism or constructivism are also proposed by the advocates of a paradigmatic renewal. Even if the solutions advocated differ, these approaches have the common characteristic of advocating a greater openness in marketing research by proposing new research methods, new forms of results, and, above all, new criteria for evaluating scientific productions.

Some limitations should be acknowledged. First, books, theses, and conference proceedings were not included in the review because it only looked at and evaluated items that were published in peer-reviewed academic journals. Future research should take this restriction into account and may decide to broaden the area of their investigations. Second, our review comprised journals identified in Scopus; upcoming studies may contemplate examining other databases. Third, the perceptions of social media practitioners may also vary by country, along with the nature of jobs and the competencies they require. To advance this research, it is

advised that our data be combined with information from various geographic contexts. Finally, the results of future research can be bed on larger sample size, as the sample in our second study was comprised of only 280 practitioners.

## Supporting information

**S1 Data.**
(XLSX)

## Author Contributions

**Methodology:** Samer Elhajjar.

**Resources:** Laurent Yacoub.

**Writing – original draft:** Laurent Yacoub.

**Writing – review & editing:** Laurent Yacoub.

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
