## [Decision Letter · Decision Letter 0]

10 Sep 2023

PONE-D-23-23751Social Media Research: We Are Publishing More but with Weak InfluencePLOS ONE

Dear Dr. Yacoub,

Thank you for submitting your manuscript to PLOS ONE. After careful consideration, we feel that it has merit but does not fully meet PLOS ONE’s publication criteria as it currently stands. Therefore, we invite you to submit a revised version of the manuscript that addresses the points raised during the review process.

We look forward to receiving your revised manuscript.

Kind regards,

Alhamzah F. Abbas, PhD

Academic Editor

PLOS ONE

Journal Requirements:

3. Please ensure that you include a title page within your main document. You should list all authors and all affiliations as per our author instructions and clearly indicate the corresponding author.

Reviewers' comments:

Reviewer's Responses to Questions

**Comments to the Author**

1. Is the manuscript technically sound, and do the data support the conclusions?

Reviewer #1: Partly

Reviewer #2: No

2. Has the statistical analysis been performed appropriately and rigorously? 

Reviewer #1: N/A

Reviewer #2: No

3. Have the authors made all data underlying the findings in their manuscript fully available?

Reviewer #1: No

Reviewer #2: No

4. Is the manuscript presented in an intelligible fashion and written in standard English?

Reviewer #1: No

Reviewer #2: Yes

5. Review Comments to the Author

Reviewer #1: I am pleased to have a chance to review this manuscript. The manuscript discusses about social media research and its gap between the theory and practice. The title and the content of the paper is interesting and informative for the academic society. However, the structure and methodology of the paper is not aligned with health research methodology. A review paper in health research should obey the principles of standard checklists of reviews like PRISMA. The authors need to address few observations noted as below:

1- The reference style must be checked according to the journal recommendations.

2- Figures should have title in main manuscript.

3- In the systematic review author did not mention the date of search and databases. If it was the same as bibliographic review, it should be mentioned.

4- The author should clarify the inclusion and exclusion criteria in systematic review.

5- In the first use of abbreviation, the full word must be written.

6- The author should use the past perfect tense when talking about actions related to conducting a study that occurred in the past and are completed.

7- The first sentence of study 1 should better be changed to:” We conducted a bibliographic study,…”

8- In study 1, the authors wrote that “we first identified three keywords related to our study” but they mentioned two key words. If the authors used the exact words, it should be in (").

9- It is better to use “excluded” instead of “dismissed” in study one, paragraph 2.

10- How was the sample size calculated in study 2 and 3?

11- In last paragraph of study one, make the following changes” VOS is superior to multidimensional scaling for constructing bibliometric analyses and maps (Van Eck and Waltman, 2009), so we did not involve multidimensional scaling.”

12- Correct the number of figure21, in last paragraph of study one.

13- Correct the number of figure 32, n study two, paragraph4.

14- Change the first sentence of fifth paragraph of study to, to following sentence: “Participants mentioned that they use the previously mentioned tools to mainly gain more information about the latest trends in the fields of social media and technology.”

15- Put table 3 and its title in right place.

16- In academic writing it is not common to use “etc”.

Reviewer #2: The manuscript entitled "Social Media Research: We Are Publishing More but with Weak Influence" aims to provide a bibliometric analysis that indicates the evolution of academic research on social media. The issue is quite interesting. Still, after reading the manuscript, I see room for improvements, which I will subsequently outline and hope that this review may help the authors as they continue to work on the paper:

• The introduction can and should be improved with paragraphs that better link the narrative from the relevance of the topic under analysis to the originality of this study. Make it clear what the originality of the study is.

• The authors said on page 8 of the file: “First, we examine the theoretical foundations of academic marketing research. Second, the research design and methodology of our three investigations are then described. Our first study involves a social media research bibliometric analysis with the goal of describing the evolution and development of academic social media research. Our second study gathers feedback and information from social media practitioners. Our third study lists suggestions for academic researchers.” Normally, I think this also depends on the field, but a good article could have only one investigation, but it should be very well done. However, I am not sure that the agglomeration of several investigations within the same study brings more value if they are not done rigorously. Thus, for instance, for bibliometric study, it is not very clear the search protocol. There are some details mentioned but not enough. Also, why the search in the database was only until March 2022? We are now in September 2023… See for instance some other bibliometric studies where the search protocol is very clearly presented:

Alshater, M. M., Atayah, F. O., & Khan, A. (2021). What do we know about business and economics research during COVID-19: A bibliometric review. Economic Research-EkonomskaIstra_zivanja, 1–29. https://doi.org/10.1080/1331677X.2021.1927786

Bota-Avram, C (2023). Bibliometric analysis of sustainable business performance: where are we going? A science map of the field. Economic Research-Ekonomska Istraživanja, 36 (1), 2137-2176. https://doi.org/10.1080/1331677X.2022.2096094

Caputo, A., Pizzi, S., Pellegrini, M., & Dabic, M. (2021). Digitalization and business models: Where are we going? A science map of the field. Journal of Business Research, 123, 489–501. https://doi.org/10.1016/j.jbusres.2020.09.053

Wang, X., Xu, Z., Su, S.-F., & Zhou, W. (2021). A comprehensive bibliometric analysis of uncertain group decision making from 1980 to 2019. Information Sciences, 547, 328–353. https://doi.org/10.1016/j.ins.2020.08.036

Wang, X., Xu, Z., Qin, Y., & Skare, M. (2021). Service networks for sustainable business: A dynamic evolution analysis over half a century. Journal of Business Research, 136, 543–557. https://doi.org/10.1016/j.jbusres.2021.07.062

• The bibliometric study is very superficial made, normally a good bibliometric study supposes to present some performance analysis and also some science mapping indicators. See for instance a very valuable article with a useful guide on how to conduct a bibliometric study. Otherwise, because the authors selected articles and analysed only the co-occurrence of the keywords from there, it is far from a valuable bibliometric analysis. Also, the authors must explain what bibliometric indicators were analysed and why?!

Donthu, N., Kumar, S., Mukherjee, D., Pandey, N., & Lim, W. M. (2021). How to conduct a bibliometric analysis: An overview and guidelines. Journal of Business Research, 133, 285–296. https://doi.org/10.1016/j.jbusres.2021.04.070

• Then, at page 12. The authors said: “In our systematic literature review…” but hold on….it is a bibliometric study or a systematic literature review? My impression is that the authors don’t know the difference between them, they think it is the same thing but definitely they are not the same. See also the article where the difference between them is explained.

• Donthu, N., Kumar, S., Mukherjee, D., Pandey, N., & Lim, W. M. (2021). How to conduct a bibliometric analysis: An overview and guidelines. Journal of Business Research, 133, 285–296. https://doi.org/10.1016/j.jbusres.2021.04.070

• However, this first study does look at all as a systematic lit review. A systematic literature review is supposed to follow a protocol (see, for instance, the PRISMA protocol)

• Also, what is the utility of bibliometric analysis if no future research directions are not identified based on bibliometric analysis??!!

• Regarding the study no.2, the authors adopt the same superficial approach. No literature review as background for study 2 is indicated. Again, it is not clear how the database with respondents was selected. How the database with 280 social media practitioners was selected? Based on what criteria? Much more rigour would be needed in an academic paper…

• In my opinion, this article is a compilation of insufficiently and non-rigorously developed studies. As a scholarship reader, I would prefer to see a single study with the necessary academic rigour rather than an agglomeration of studies insufficiently developed within the same article.

• As a value-added bibliometric analysis, I was expected to see at least a brief discussion on the most popular topics ( a content analysis), from where a useful summary of future research directions could be developed for the researchers interested in this field. In addition, a discussion of the emerging trends in research for the field under investigation would be very useful for the readers.

FINAL COMMENT

I know the effort it takes to develop rigorous, serious, and useful research work. The comments I have just made are intended to help you better your scientific work and reflect only my point of view. I hope these comments and suggestions can help improve this paper.

Good work and good luck with this journey!

6. PLOS authors have the option to publish the peer review history of their article (what does this mean?). If published, this will include your full peer review and any attached files.

Reviewer #1: **Yes: **Zahra Ghorbani

Reviewer #2: No

---

## [Author Response · Author response to Decision Letter 0]

4 Nov 2023

We are very grateful to the editorial office for the comments and thoughtful suggestions. Based on these comments and suggestions, we have made careful modifications to the original manuscript. 

In an effort to facilitate clarity and ease of identification, all the new changes and adjustments have been highlighted in blue in the revised manuscript.

We have completed the necessary adjustments to the reference style in accordance with the journal's recommendations, as per your feedback. We appreciate your diligence in pointing out this issue and providing guidance for improvement.

We have made the necessary revisions to include titles for the figures in the main manuscript, as per your feedback.

We have addressed the issue regarding the omission of the search date, as per your feedback.

We clarified the inclusion and exclusion criteria in systematic review. 

We have ensured that the full word is written out upon the initial use of each abbreviation.

We made the necessary adjustments to ensure that the past perfect tense is correctly used when describing actions related to conducting the study that occurred in the past and are completed. This modification enhances the clarity and accuracy of the narrative.

---

## [Decision Letter · Decision Letter 1]

2 Jan 2024

Social Media Research: We Are Publishing More but with Weak Influence

PONE-D-23-23751R1

Dear Dr. Laurent Yacoub,

We’re pleased to inform you that your manuscript has been judged scientifically suitable for publication and will be formally accepted for publication once it meets all outstanding technical requirements.

Kind regards,

Alhamzah F. Abbas, PhD

Academic Editor

PLOS ONE

Additional Editor Comments (optional):

Reviewers' comments:

Reviewer's Responses to Questions

**Comments to the Author**

1. If the authors have adequately addressed your comments raised in a previous round of review and you feel that this manuscript is now acceptable for publication, you may indicate that here to bypass the “Comments to the Author” section, enter your conflict of interest statement in the “Confidential to Editor” section, and submit your "Accept" recommendation.

Reviewer #3: All comments have been addressed

2. Is the manuscript technically sound, and do the data support the conclusions?

Reviewer #3: Yes

3. Has the statistical analysis been performed appropriately and rigorously? 

Reviewer #3: N/A

4. Have the authors made all data underlying the findings in their manuscript fully available?

Reviewer #3: Yes

5. Is the manuscript presented in an intelligible fashion and written in standard English?

Reviewer #3: Yes

6. Review Comments to the Author

Reviewer #3: The revised version of this article is much better than the previous version. Many changes or additions suggested in the review have been taken into consideration by authors and the paper has been improved now in the visible way. Moreover, the authors explained all improvements made in details and I am personally happy with the updated version.

7. PLOS authors have the option to publish the peer review history of their article (what does this mean?). If published, this will include your full peer review and any attached files.

Reviewer #3: No

---

## [Editor Report · Acceptance letter]

30 Jan 2024

PONE-D-23-23751R1 

PLOS ONE

Dear Dr. Yacoub, 

I'm pleased to inform you that your manuscript has been deemed suitable for publication in PLOS ONE. Congratulations! Your manuscript is now being handed over to our production team.

Kind regards, 

on behalf of

Dr. Alhamzah F. Abbas 

Academic Editor

PLOS ONE